# Near-Infrared Artificial Optical Synapse Based on the P(VDF-TrFE)-Coated InAs Nanowire Field-Effect Transistor

**DOI:** 10.3390/ma15228247

**Published:** 2022-11-21

**Authors:** Rui Shen, Yifan Jiang, Zhiwei Li, Jiamin Tian, Shuo Li, Tong Li, Qing Chen

**Affiliations:** 1Key Laboratory for the Physics and Chemistry of Nanodevices, School of Electronics, Peking University, Beijing 100871, China; 2Academy for Advanced Interdisciplinary Studies, Peking University, Beijing 100871, China

**Keywords:** optical synapse, near-infrared, InAs nanowire (NW), field-effect transistors (FET), P(VDF-TrFE)

## Abstract

Optical synapse is the basic component for optical neuromorphic computing and is attracting great attention, mainly due to its great potential in many fields, such as image recognition, artificial intelligence and artificial visual perception systems. However, optical synapse with infrared (IR) response has rarely been reported. InAs nanowires (NWs) have a direct narrow bandgap and a large surface to volume ratio, making them a promising material for IR detection. Here, we demonstrate a near-infrared (NIR) (750 to 1550 nm) optical synapse for the first time based on a poly(vinylidene fluoride-trifluoroethylene) (P(VDF-TrFE))-coated InAs NW field-effect transistor (FET). The responsivity of the P(VDF-TrFE)-coated InAs NW FET reaches 839.3 A/W under 750 nm laser illumination, demonstrating the advantage of P(VDF-TrFE) coverage. The P(VDF-TrFE)-coated InAs NW device exhibits optical synaptic behaviors in response to NIR light pulses, including excitatory postsynaptic current (EPSC), paired-pulse facilitation (PPF) and a transformation from short-term plasticity (STP) to long-term plasticity (LTP). The working mechanism is attributed to the polarization effect in the ferroelectric P(VDF-TrFE) layer, which dominates the trapping and de-trapping characteristics of photogenerated holes. These findings have significant implications for the development of artificial neural networks.

## 1. Introduction

With the continuous advancement of semiconducting technology, traditional von Neumann architectures are confronting extreme bottlenecks in processing speed and energy consumption [1,2]. Neuromorphic computing is a newly computing architecture attempting to overcome the limitations of von Neuman-based computing [3,4]. Light has been widely used as a communication medium in telecoms and data centers. Infrared (IR) light detection is especially important in biomedical imaging, communications engineering and machine vision [5,6,7]. However, few materials can achieve wide-spectrum infrared detection because many materials have large band gaps. Besides, light is expected to play a significant part in high-speed real-time artificial neuromorphic computing with low latency, low power consumption, and high bandwidth [8]. Various optical synaptic devices have been developed as essential components of optical neuromorphic systems [9]. Controlling the relaxation properties of photogenerated carriers is crucial for the realization of optical synapses, since the photocurrent can be sustained for a long period after illumination, thereby providing memory features for the simulation of synapse functions [10,11]. To this end, many optical synaptic devices have been designed on different principles, including those based on phase-change materials [12,13], memristors [14,15], ferroelectrics [16,17,18,19], charge trapping/de-trapping [20,21,22,23] or ion transport mechanisms [24].

Synaptic behaviors under light stimulation have been simulated using various materials, such as nanowires (NWs) [25,26], Si nanocrystals (NCs) [27,28], oxide semiconductors [29,30], two-dimensional materials [17,31,32,33,34,35], indium arsenide (InAs) dots [36], hybrid perovskites [37,38], and organic materials [19,39,40,41]. The basic synaptic functions, such as excitatory postsynaptic current (EPSC)/inhibitory postsynaptic current (IPSC), paired-pulse facilitation (PPF), and short-term plasticity (STP)/long-term plasticity (LTP), have been simulated using light stimuli [9,42]. Unfortunately, most of the current optical synaptic devices can only work in the ultraviolet-visible light region, while the IR optical synaptic device has rarely been reported based on organic/inorganic materials. InAs NWs have a direct narrow bandgap (~0.35 eV) naturally providing significant advantages in IR detection [43,44]. Although InAs NW optical synaptic devices have been demonstrated in the UV and visible light region [21,45], there have been no reports on InAs NW optical synapse in IR region.

Here, we present an InAs NW IR optical synaptic device in the near-infrared (NIR) range (750 to 1550 nm) for the first time with the help of ferroelectric material poly(vinylidene fluoride-trifluoroethylene) (P(VDF-TrFE)) coverage. The functions of optical synaptic devices, such as EPSC, PPF, and STP/LTP, are demonstrated by light stimulus.

## 2. Materials and Methods

### 2.1. Nanowire Growth and Device Fabrication

High-quality single-crystalline InAs NWs were synthesized by metal–organic chemical vapor deposition (MOCVD). The detailed growth process and characterization have been reported in our previous work [46]. The as-grown InAs NWs were transferred and uniformly dispersed onto a heavily p-doped Si substrate covered by a 300 nm-thick SiO_2_. After the electrodes were patterned by electron-beam lithography, the surface oxide layer outside InAs NW in the contact area was removed by (NH_4_)S_x_ solution before the metal deposition to ensure ohmic contact between the metal electrodes and InAs NW. A double-layer metal Ti/Au with a thickness of 5/90 nm was deposited on the surface of InAs NW by electron beam evaporation and liftoff. P(VDF-TrFE) (70:30 in mol%) thin films were spin-coated on the channel of the device and then annealed at 130 °C for 30 min to crystallize. The detailed device fabrication process is the same as that in the previous reports by our group [46].

### 2.2. Characterization and Measurement

High-resolution transmission electron microscopy (HRTEM) is used to characterize the lattice quality of InAs NW in a Tecnai F20 TEM (FEI, Portland, OR, USA). The morphology of InAs NWs on the transferred substrate are observed and positioned by scanning electron microscopy (SEM) (using a Quanta FEG 600, FEI, Brno, Czech Republic), and the diameter of the NW is measured by atomic force microscopy (AFM) (using a Multimode 8 system, Bruker, Karlsruhe, Germany). All optoelectrical measurements are carried out in air at room temperature on a LabRAM HR800 evolution system (Horiba Jobin-Yvon, Paris, France) with NKT EXR-15 supercontinuum laser sources. Simultaneous optoelectrical characterization was performed using a Keithley 4200 semiconductor analyzer (Keithley, Cleveland, OH, USA) equipped with a probe station. The detailed optoelectrical measurements can be found in the previous report by our group [47].

## 3. Results and Discussion

### 3.1. The Pristine InAs NW Device Optoelectronic Properties

Figure 1a is a transmission electron microscopy (TEM) image of the NWs, indicating that the NWs have smooth surface and uniform diameter. Figure 1b,c are the high-resolution TEM (HRTEM) image and electron diffraction pattern, respectively, showing the NW has a wurtzite (WZ) structure with the axis along the [0001] direction and is covered by a 2.5 nm native oxide layer due to the exposure to air after the growth. Before depositing the Ti/Au = 5/90 nm electrodes, the oxide layer in the contact area was etched by (NH_4_)S_x_ solution for 90 s to ensure ohmic contact. The scanning electron microscopy (SEM) image of a typical InAs NW field-effect transistor (FET) is shown in Figure 2a. The channel length of the device is 1.2 μm, and the diameter of the NW (with the native oxide layer) is measured to be 23 nm by atomic force microscopy (AFM), as shown in Figure 1d.

Before investigating the optoelectronic properties of the device, the electrical properties of the InAs NW FETs are measured. The p-type heavily doped Si substrate is used as the back-gate, and the 300 nm SiO_2_ layer is used as the back-gate dielectric layer. The output and transfer characteristic curves of the as-fabricated back-gate device (the pristine device) are measured at room temperature. The output characteristics are shown in Figure 3a. The source-drain current (*I*_ds_) increases linearly with the source-drain voltage (*V*_ds_) at the beginning and tends to saturate at large *V*_ds_ for different gate voltages (*V*_gs_). Such phenomena confirm good ohmic contact in the device due to the Fermi level pinning near the bottom of the conductance band of InAs [48]. The transfer curve of the device is shown in Figure 3b. During the measurement, a scanning voltage of −10 V to 10 V is applied to the gate, and a fixed bias voltage *V*_ds_ of 0.1 V is applied between the source and drain electrodes. The device shows an n-type characteristic with a high on/off ratio of ≈10^5^ in a *V*_gs_ sweeping range of −10 V to 10 V, as shown in Figure 3b. The field-effect mobility (μ) can be extracted using the following equation [49,50]:(1)$$\mu =\frac{gL_{g}^{2}}{C_{\mathrm{ox}}V_{\mathrm{ds}}}$$
where *g* is the transconductance (calculated from the measured transfer curve using g = d*I*_ds_/d*V*_gs_), *L*_g_ is the gate length, $C_{\mathrm{OX}}$ is the gate oxide capacitance of the devices and can be calculated using $C_{\mathrm{OX}}=2\pi \varepsilon_{0}\varepsilon_{r}/\cos h\left[\left(t_{\mathrm{ox}}+r_{\mathrm{NW}}\right)/r_{\mathrm{NW}}\right]$, where $\varepsilon_{0}$ is the vacuum permittivity, $\varepsilon_{r}$ is the relative permittivity of the dielectric, $t_{\mathrm{ox}}$ is the thickness of the gate oxide layer, and $r_{\mathrm{NW}}$ is the radius of InAs NW. The field-effect mobility is calculated to be 1116 cm^2^/V·s from the measured transfer curve, which is comparable to the previously reported field-effect mobilities in InAs NW FETs at room temperature [51].

The optoelectronic properties of InAs NW FETs are measured in air at room temperature and at *V*_gs_ = 0 V unless otherwise specified. As shown in the inset of Figure 2b, the dark current *I*_dark_, referring to the *I*_ds_ without light illumination, has an ultralow value of 0.1 nA at *V*_ds_ = 0.1 V, which is close to that of the latest reported NW-based advanced photodetector [52,53]. A positive photoconductive effect with highly sensitive photodetection is observed under 750 nm light illumination. Even when the light intensities are as low as 0.1 W/cm^2^, the current of the device under 750 nm laser illumination is greater than that in the dark. Figure 2b displays the current of the pristine device increases from 0.18 nA to 0.6 μA at *V*_ds_ = 0.1 V as the intensity of the 750 nm light increases from 0.1 to 2381.4 W/cm^2^. The current on/off ratio of the device is about 6 × 10^3^ for the light with the intensity of 2381.4 W/cm^2^ at *V*_ds_ = 0.1 V. In Figure 2c, the photocurrent (*I*_photo_, defined as the difference of the current under light illumination and in the dark) at *V*_ds_ = 0.1 V increases nonlinearly with the incident light power. In previous works, the relationship between experimental photocurrent and incident light power has been fitted with a power function, which reflects the generation mechanism of photocurrent under light illumination [54]. Hence, we also fit the experimental data using the equation of *I*_photo_~*P^m^*, where *m* is the exponent of the power function and *P* is the incident light power, respectively. When *m* approaches 1, it means that the generation-recombination process is the main origin of the photocurrent. However, a value of *m* less than 1 is generally observed, which indicates that the photoelectric conversion process contains complex mechanisms, including electron-hole generation, recombination and trapping [55]. Here, in the present InAs NW optical synapse, the fitting value *m* is 0.81, which indicates that the defect trapping process contributes to the photocurrent a lot. Figure 2d shows the time-resolved current response to the 750 nm light pulses with different light intensities. It shows clearly that after the light is switched off, the current quickly returns to the initial current value, indicating the pristine InAs NW device does not have synaptic characteristics.

### 3.2. The P(VDF-TrFE)-Coated InAs NW Device Optoelectronic Properties

Recently, we have found that ferroelectric polymer P(VDF-TrFE) coating not only significantly improves the electrical performance of InAs NW FETs but also enables a very good electrical synaptic behavior of the devices [56]. We therefore study here whether the P(VDF-TrFE) coating can improve the optoelectronic performance of the device or even induce optical synaptic behaviors in the InAs NW FETs. As can be seen in Figure 4, the output characteristics of the P(VDF-TrFE)-coated InAs NW FET display an on-state current of 14.2 μA at *V*_ds_ = 0.5 V, which is much larger than that in the pristine device. Figure 5a shows the transfer curves of the device after the P(VDF-TrFE) coating in the dark and under the illumination of 750 nm laser. The transfer curves do not coincide when the gate voltage scans from −10 V to 10 V and from 10 V to −10 V, as shown in Figure 5a, indicating that there is a hysteresis in the transfer curves. The hysteresis, corresponding to the storage window, increases as the light intensity increases, also shown in Figure 5a. We use the threshold voltage, *V*_T_, to quantitatively describe the hysteresis in the transfer curves. The *V*_T_ is measured by elongating the tangent line of the transfer curve in linear scale to intersect with the horizontal axis, the intersection point is the *V*_T_ value, as shown in Figure 5b. It can be seen in Figure 5b, that the threshold voltage (*V*_T_) moves to the left under the light illumination for *V*_gs_ sweeping from −10 V to 10 V, and the higher the light intensity, the more the movement. As shown in Figure 5a the transfer curves do not move obviously when *V*_gs_ is swept from 10 V to −10 V. Therefore, the hysteresis increases with the light intensity under the illumination of 750 nm light.

After the P(VDF-TrFE) coating, the dark current at *V*_gs_ = 0 V increases to 9.6 nA and the photocurrent increases more obviously than that in the pristine device with the same light intensity, as shown in Figure 5c. The P(VDF-TrFE) coating not only reduces surface defects but also enhances the electric field around the NW and increases the performance of the P(VDF-TrFE)-coated InAs NW FET [56], which explains the increase of the dark current. The responsivity (*R*) of a photodetector is widely used to evaluate the sensitivity of photodetectors and can be calculated as: $R=\frac{I_{\mathrm{photo}}}{P\times A}$, where *P* is the incident light intensity and *A* (=L×d, where *L* is the channel length, *d* is the NW diameter) is the effective irradiation area on the NW [57]. Here, from the experimental values of photocurrent *I*_photo_ under different light intensities extracted from Figure 2b and Figure 5c, the responsivities are obtained for the devices before and after P(VDF-TrFE) coating. Figure 5d compares the responsivity of the device with and without P(VDF-TrFE) coating for different light intensities. It shows that the responsivity of the device after P(VDF-TrFE) coating is greater than that of the pristine device for all the light intensities, demonstrating the benefit of P(VDF-TrFE) coating on improving the photoelectronic performance of the InAs NW devices. The responsivity of the P(VDF-TrFE)-coated device decreases with the increase in the light intensity. The maximum responsivity is obtained to be 839.3 A/W at the lowest light intensity of 0.1 W/cm^2^ of 750 nm light.

### 3.3. Optical Synaptic Behavior Based on P(VDF-TrFE)-Coated InAs NW FET

Biological synapses are the basic units for learning, memory, and information processing by human beings. As schematically shown in Figure 6a, the neurotransmitters released by the pre-synapse act on the post-synapse and cause the changes in post-synapse potential in response to various external stimuli [32,58]. To mimic the response of biological synapses to light stimulation, the P(VDF-TrFE)-coated InAs NW FET is regarded as a synapse with light pulses as the presynaptic stimulus. The source and drain electrodes act as the post-synapse. The source-drain current *I*_ds_ is the EPSC and the channel conductance represents the synaptic weight. Figure 6b shows the current change in a P(VDF-TrFE)-coated InAs NW device after applying individual 750 nm light pulse with different light intensities at *V*_ds_ = 0.1 V. It displays clearly that the current increases immediately when the light stimuli arrives and then decreases slowly after the light is switched off. The current just after the light pulses (that is EPSC) and the recovery time (the time needed for the current to return to the original dark current value after the light is switched off) increase with the increasing light intensity, exhibiting a typical EPSC behavior similar to a biological synapse. Additionally, when the duration time of the light pulse increases from 0.5 s to 2.0 s, EPSC increases from 0.27 μA to 0.38 μA and the current recovery time also increases, as shown in Figure 6c. Therefore, increasing the intensity or the duration of the light pulse increases both the amplitude of EPSC and the recovery time, successfully simulating a transformation from STP to LTP in biological synapses.

Due to a short-term memory (STM) effect, when a synaptic device is stimulated by two consecutive stimuli, the post-synaptic current caused the second stimulus will be higher than that by the first stimulus, as shown in Figure 6d. Paired-pulse facilitation (PPF) refers to the increases in the ratio between the two post-synaptic currents caused by two consecutive stimuli [21], showing a STM capacity of the synaptic device. It can be seen from Figure 6d that the EPSC after the second light pulse (*A*_2_, 0.12 μA) is greater than that after the first light pulse (*A*_1_, 0.05 μA) with the same light intensity of 26.7 W/cm^2^ and the same duration time of 0.2 s, demonstrating a typical PPF behavior. The mechanism of PPF in our device could be that electron-hole pairs are generated in the InAs NW channel by the light illumination. After the first light pulse, some of the photo-generated holes are trapped at the InAs/InO_X_ interface and/or in the InO_X_ layer. If the interval time is short enough, the trapped holes generated during the first light pulse will not be completely released before the second light pulse arrives. Correspondingly, the movable electrons in the InAs NW channel induced by the trapped holes will not be recombined completely, so that the photogenerated carriers generated by the first light pulse do not decay completely. At this moment, the second light pulse arrives and generates more electron-hole pairs. Thus, the post-synaptic current that triggered by the second 750 nm light pulse is larger than the first one. With a longer pulse interval, more trapped holes are released, resulting in a smaller second post-synaptic current. 

PPF index, which is defined as the ratio of $\frac{A_{2}-A_{1}}{A_{1}}\times 100\%$, is used to quantitatively indicate PPF behavior. *A*_1_ and *A*_2_ are the EPSCs for the first and the second light pulses, respectively, as shown in Figure 6d. In Figure 6e, PPF indices are summarized as the interval time of the two light pulses (with light wavelength of 750 nm, light intensity of 26.7 W/cm^2^ and duration time of 0.2 s). Part of the original data for Figure 6e are displayed in Figure 7. PPF index is 160% when the interval time is 0.16 s and decreases rapidly with the interval time increasing, suggesting that memory-loss accelerates with increasing interval time. The relationship between PPF index and interval time has been reported to be fitted by a double exponential function of the interval time [42]:(2)$$\mathrm{PPF}\;\mathrm{index}=c_{1}e^{-\frac{\Delta t}{\tau_{1}}}+c_{2}e^{-\frac{\Delta t}{\tau_{2}}}+c_{0}$$
where Δt is the interval time of the two consecutive light pulses, $c_{0}$, $c_{1}$ and $c_{2}$ are characteristic constants, $\tau_{1}$ and $\tau_{2}$ correspond to the characteristic time of the fast phase and slow phase, respectively. Through fitting the experimental data in Figure 6e by a double exponential function, two-time constants are obtained to be $\tau_{1}=$ 0.02 s and $\tau_{2}=$ 0.23 s, which are similar to those in biological synapse. The values of $\tau_{2}$ are about one order of magnitude larger than those of $\tau_{1}$ for the incident laser wavelength of 750 nm, consistent with the decay characteristics of the PPF index in biological synapses [42,59]. The decay in PPF index with interval time could be understood as followed. When the interval time Δt is apparently smaller than the EPSC decay time, PPF index is rather sensitive to Δt. PPF index sharply decreases with the increase of Δt, giving rise to the rapid decay process of PPF index with the characteristic relaxation time of $\tau_{1}$. When Δt is longer than the EPSC decay time, the enhancement of the photocurrent stimulated by the second spike is weakened, so that PPF index decays slowly with the characteristic relaxation time of $\tau_{2}$.

In the human body, synaptic plasticity is also closely related to the process of memory formation. STM, which is stored in the hippocampus and easily annihilated by new information, can be consolidated and transformed into long-term memory (LTM) through sustained repetitive practice [60]. Figure 6f shows the EPSCs response to 20 light pulses stimulus (0.5 s duration, 50 ms interval time) of the P(VDF-TrFE)-coated InAs NW device with different light intensities of 750 nm laser. When the light pulses with an intensity of 2.9 W/cm^2^ are applied, EPSC is 0.2 μA after 20 light pulses, and then decreased slowly. With the increasing number of the light pulses, EPSC increases sequentially from 0.1 μA (after the first light pulse) to 1.8 μA (after 20 light pulses) with light intensity of 26.7 W/cm^2^. These results confirm that increasing the number or intensity of light pulse stimulation induces larger EPSC, thereby exhibiting a transition from short-term to long-term plasticity in the P(VDF-TrFE)-coated InAs NW optical synapse.

Furthermore, synaptic responses of the P(VDF-TrFE)-coated InAs NW device to the light with other wavelengths in the NIR range are also studied. Figure 8a shows that the P(VDF-TrFE)-coated InAs NW device responds to the lights with wavelength up to 1550 nm, which is of great significance in communication applications. When the light wavelength increases from 900 nm to 1550 nm, EPSC of the P(VDF-TrFE)-coated InAs NW device decreases from 0.3 μA to 0.06 μA and a shorter time is needed to recover to the initial current value after the light pulses, showing a decrease in synaptic response with light wavelength increasing. To further mimic multilevel states of conductance under light stimulation, 20 light pulses with different wavelengths (with the same light intensity of 26.7 W/cm^2^, duration time of 0.5 s and interval time of 50 ms) are continuously applied to gradually increase the EPSCs in Figure 8b. When the light wavelength changes from 900 nm to 1550 nm, after applying 20 light pulses, the current change rate (defined as the ratio of the current at 60 s to that before applying the light pulses) decreases from 335% to 169%. Also, the recovery time is longer when the wavelength is shorter. The above results indicate that the longer the wavelength, the weaker the performance of the optical synapses. The present results reveal that the variation in light wavelength or intensity can induce different response characteristics of the synapses, so that the present optical synapses have a large dynamic range and advanced resolution of IR light with different colors and intensities for image recognition. 

So far, NIR optical synapses are still rare. In Table 1 we summarize all the reports on NIR optical synapses in the literature [1,2,3,4,5,6,7,8,9,10,11,12,13,14,15,16,17,18,19,23,27,28,30,35,38,39,40,41,61,62,63,64] as far as we know. It shows that none of the organic devices and only a couple of inorganic devices can respond to NIR light with a wavelength longer than 1500 nm. Among the inorganic devices, all the previously reported NIR optical synapses are based on nanocrystals or two-dimensional materials. Our device is the first reported NIR optical synapse based on nanowires.

Several mechanisms have been suggested previously for InAs NW synapses, such as charge trapping and de-trapping at the InAs/Al_2_O_3_ interface, light-induced photogating effect or negative photoconductive effect at the NW surface [21,65]. However, we did not observe optical synaptic behavior in NIR region in the pristine InAs NW FET. The reasons could be that our NWs and devices have high quality and there is not a high-k layer covering on the pristine InAs NW FETs. Anyhow, the previously suggested mechanisms cannot explain the optical synaptic behavior displayed in the present work. Other studies have shown that the relaxation properties of photogenerated carriers can be controlled by different polarizations of the ferroelectric layer, resulting in the change in conductance and optical synaptic weight updating [16,66,67]. In order to explore the working mechanism of the P(VDF-TrFE)-coated InAs NW optical synapse, we study the optical synaptic behaviors under different polarizations of P(VDF-TrFE).

Different *V*_gs_ pulses are applied to the back-gate electrode to manipulate the polarization of P(VDF-TrFE). As shown in Figure 9, after applying 10 negative (positive) *V*_gs_ pulses (the duration time and interval time are both 50 ms), the current at *V*_ds_ = 0.1 V is higher (lower) than that before the *V*_gs_ pulses due to the downward (upward) polarization of P(VDF-TrFE). The residual polarization in the P(VDF-TrFE) layer leads to the energy band of InAs NW bending near the surface, thus resulting in the accumulation (or depletion) of electrons near the surface.

Next, we study the synaptic performance of the P(VDF-TrFE)-coated InAs NW devices under different polarization condition. Figure 10a shows the EPSCs of the P(VDF-TrFE)-coated InAs NW device response to 10 consecutive light pulses (with wavelength being 750 nm, pulse duration time being 0.5 s, interval time being 0.5 s) before P(VDF-TrFE) polarization. The results are similar to those shown in Figure 6f. Then, the P(VDF-TrFE) layer is polarized by applying 10 positive or negative *V*_gs_ pulses (with the same voltage amplitude of 5 V, duration time being 50 ms and interval time being 50 ms). It is worth noting that before measuring the time-resolved photo-response, the current needs to be completely stabilized after applying the *V*_gs_ pulses to ensure that P(VDF-TrFE) maintain stable residual polarization for analysis, as shown in Figure 9. For comparison, the light pulses used in Figure 10a,c,e are the same. Figure 10c shows the EPSCs to 10 light pulses after applying 10 negative *V*_gs_ pulses, where no obvious synaptic response can be observed. Then, the P(VDF-TrFE) is polarized to the upward direction by applying 10 positive *V*_gs_ pulses. Figure 10e shows the EPSCs to 10 light pulses after applying 10 positive *V*_gs_ pulses, the synaptic performance is even better than that before applying *V*_gs_ pulses. The change rate of *I*_ds_ is calculated as the ratio of the current after the tenth light pulse and the initial current before applying the first light pulse, that is, the change in synaptic weight, which reflects the information processing ability of synaptic connection. EPSC slowly decreases and remains ≈83% larger than the initial current after 30 s in the upward polarization condition, which is larger than the unpolarized condition (≈29%) in Figure 10a.

Based on the above experimental results, we propose that the working mechanism of the P(VDF-TrFE)-coated InAs NW device is due to the polarization effect in the ferroelectric P(VDF-TrFE) layer, which controls the trapping and de-trapping characteristics of photogenerated holes. For the P(VDF-TrFE)-coated InAs NW device before applying *V*_gs_ pulses, spontaneous ferroelectric polarization exists, so the energy level of the hole traps near the InAs NW surface is higher than the valence band energy (*E*_v_) of InAs NW, as illustrated in Figure 10b. Under light illumination, part of the photo-generated holes tends to be trapped in the hole-traps, resulting in the increase of electron densities in the NW channel, and the current increases. After the light pulse, the trapped holes need some time to be de-trapped, the current decreases to the original value slowly, which is ≈29% larger than the initial current after 30 s, corresponding to the optical synaptic behavior shown in Figure 6 and Figure 10a. After applying negative *V*_gs_ pulses, the energy bands of InAs NW near the InAs/InO_x_ interface bend downward, shifting down the hole-trap energy level near the InAs NW surface, as shown in Figure 10d. Under the illumination of light pulses, the photogenerated carriers can rarely be captured and the current rapidly returns to the initial level, so that no obvious optical synaptic property is shown in Figure 10c. In contrast, after applying positive *V*_gs_ pulses, the energy band of InAs NW near the InAs/InO_x_ interface bends upward, shifting up the hole-trap energy level, as shown in Figure 10f. The rise of the hole-trap energy level is beneficial to the trapping of photogenerated holes near the surface of the NWs. In Figure 10e, after 10 light pulses illumination, EPSC slowly decreases and remains ≈83% larger than the initial current after 30 s, which is larger than the initial state (≈29%) shown in Figure 10a. As a result, a remarkable optical synaptic behavior is observed for the P(VDF-TrFE)-coated InAs NW FET in Figure 10e. Therefore, the optical synaptic property in the present P(VDF-TrFE)-coated InAs NW FET is attributed to the polarization in P(VDF-TrFE), which results in photogenerated holes trapping near the surface of InAs NW.

## 4. Conclusions

In conclusion, we demonstrate an optical synapse for NIR light ranging from 750 nm to 1550 nm using a P(VDF-TrFE)-coated InAs NW FET. The photocurrent and responsivity of the device both increase after P(VDF-TrFE) coating, indicating the advantages of P(VDF-TrFE) coverage for improving the photoelectric performance of InAs NW device. Importantly, the P(VDF-TrFE)-coated InAs NW FET exhibits optical synaptic properties in response to NIR lights ranging from 750 nm to 1550 nm. EPSC, PPF, and a STP to LTP transition are demonstrated by the P(VDF-TrFE)-coated InAs NW FET through changing the wavelength, intensity, pulse duration time or pulse number of the incident light. Moreover, the P(VDF-TrFE)-coated InAs NW FET displays different responses to NIR light after applying gate voltage pulses. The working mechanism is attributed to the polarization effect in the ferroelectric P(VDF-TrFE) layer, which controls the trapping and de-trapping characteristic of photogenerated holes. The present findings demonstrate the great potential of InAs NW-based artificial synapses in high-performance optical synapses, paving the way for the construction of all-optical neuromorphology networks.

## Figures and Tables

**Figure 1 materials-15-08247-f001:**
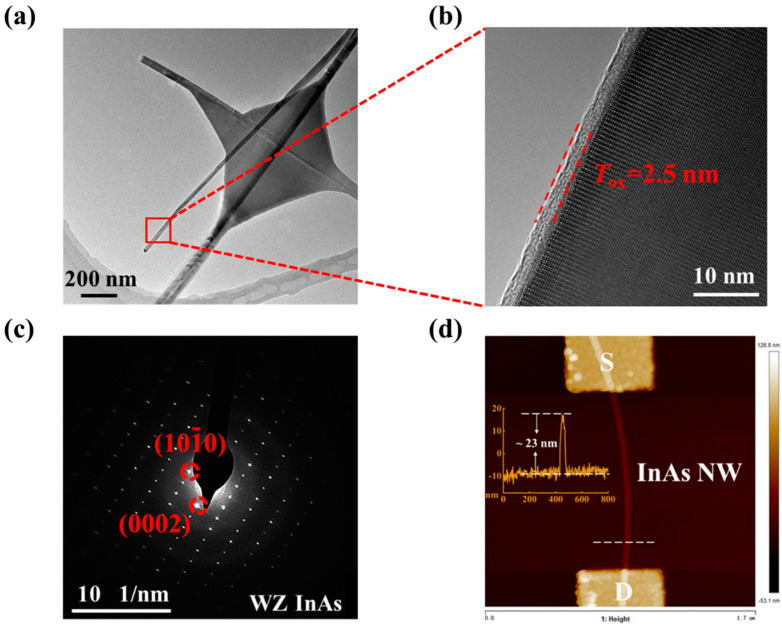
(**a**) Transmission electron microscopy (TEM) image of the as-grown InAs NWs. (**b**) High-resolution transmission electron microscopy (HRTEM) image of the InAs NW outlined in (**a**) by a red square. The thickness of the native oxide layer is 2.5 nm, as marked by the dashed lines. (**c**) Electron diffraction pattern of the NW shown in (**a**,**b**). The pattern is indexed to be [010] zone axis and the growth direction of the InAs NW is [0001],as marked by the dashed circle. (**d**) Atomic force microscopy (AFM) image of an InAs NW field-effect transistor, where the diameter of the NW is measured to be 23 nm.

**Figure 2 materials-15-08247-f002:**
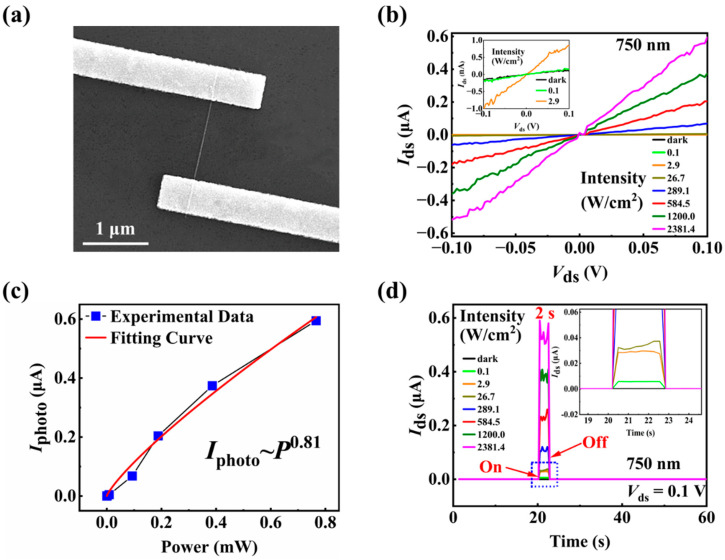
(**a**) Scanning electron microscopy (SEM) image of a typical InAs NW FET. (**b**) *I*_ds_-*V*_ds_ curves in the dark and under 750 nm light illumination with different light intensities. The inset is the enlarged image showing the data with low current. (**c**) The relationship between photocurrent and incident light power. (**d**) The time-resolved current response to 750 nm light pulses with 2 s duration time and different light intensities. The inset is the enlarged image showing the data with low current.

**Figure 3 materials-15-08247-f003:**
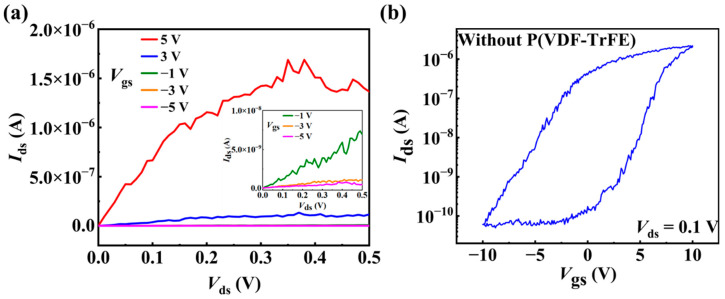
Electronical characteristics of the as-fabricated back-gate device (the pristine device). (**a**) The output characteristic curves measured under different gate voltages. The inset shows the data at low current level. (**b**) The transfer characteristic curve measured at *V*_ds_ = 0.1 V.

**Figure 4 materials-15-08247-f004:**
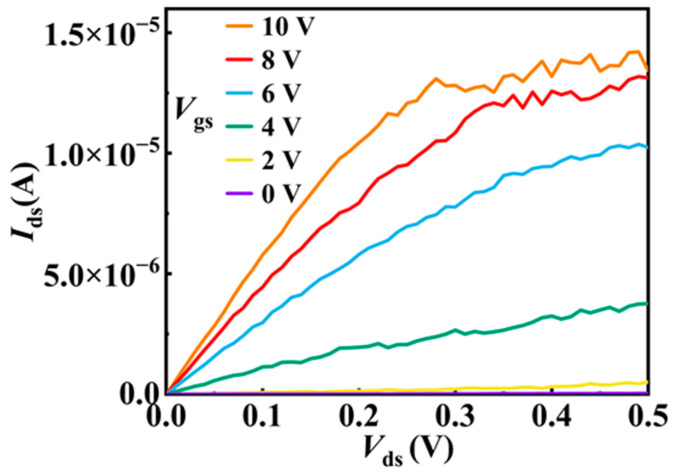
The output characteristic curves of the P(VDF-TrFE)-coated InAs NW device.

**Figure 5 materials-15-08247-f005:**
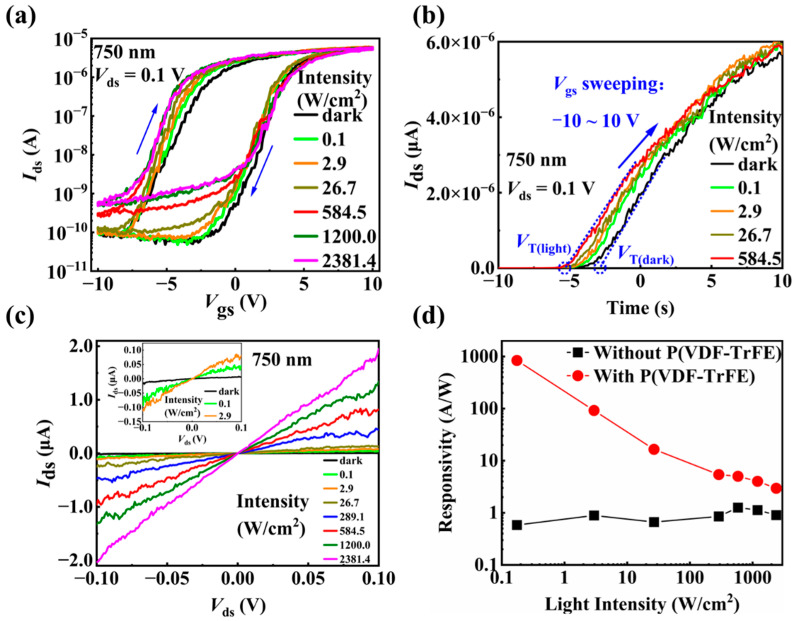
Optoelectronic properties of the P(VDF-TrFE)-coated InAs NW device. (**a**) The transfer characteristic curves measured in the dark and under light illumination with the light intensities increasing from 0.1 to 2381.4 W/cm^2^ at *V*_ds_ = 0.1 V. (**b**) The transfer characteristic curves in linear coordinates showing the shift of the threshold voltages (*V*_T_) in the dark and under light illumination with different intensities. (**c**) *I*_ds_-*V*_ds_ curves in the dark and under 750 nm light illumination with different intensities. The inset is the enlarged image showing the data with low current. (**d**) The responsivity (*R*) at different light intensities of the devices with/without P(VDF-TrFE) coating.

**Figure 6 materials-15-08247-f006:**
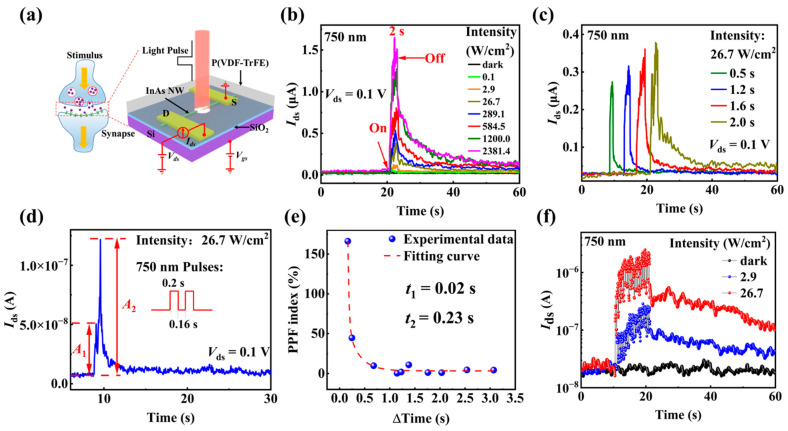
(**a**) Schematic diagram of a biological synapse and the P(VDF-TrFE)-coated InAs NW optical synaptic device. (**b**) The EPSC synaptic response to individual 750 nm laser pulses (duration time being 2 s) with different light intensities. The arrows in the figure indicate the time when the light is turned on and off. (**c**) The EPSC synaptic response to different duration time with 750 nm laser pulse stimuli. (**d**) PPF effect stimulated by paired light pulses with duration being 0.2 s and interval time being 0.16 s. The lengthes indicated by the arrows represent the change of current. (**e**) PPF index as a function of the light pulse interval time (0~3.3 s). Fitting the data with a double exponential function gives two-time constants of 0.02 s and 0.23 s. (**f**) The EPSCs of the P(VDF-TrFE)-coated InAs NW FET under 20 light pulse stimuli with wavelength being 750 nm.

**Figure 7 materials-15-08247-f007:**
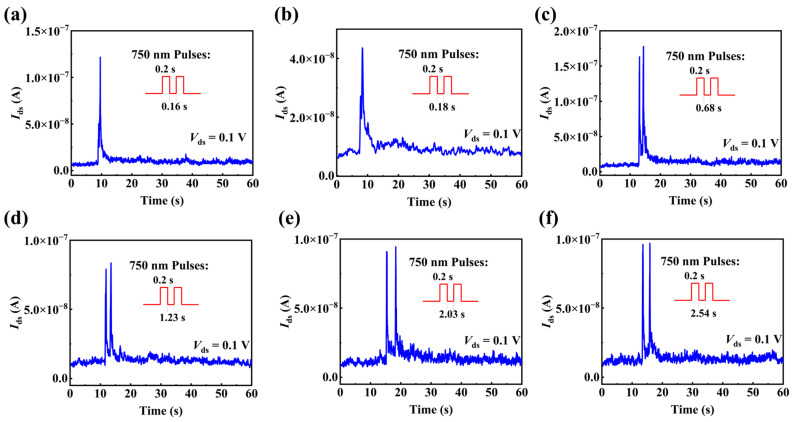
The post-synaptic current (PSC) of the P(VDF-TrFE)-coated InAs NW optical synapse stimulated by paired light pulses with the wavelength of 750 nm, intensity of 26.7 W/cm^2^, duration time of 0.2 s and different interval time (**a**) 0.16 s, (**b**) 0.18 s, (**c**) 0.68 s, (**d**) 1.23 s, (**e**) 2.03 s, (**f**) 2.54 s.

**Figure 8 materials-15-08247-f008:**
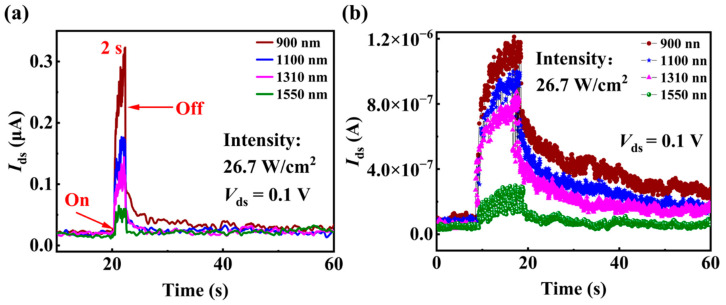
(**a**) The EPSCs response to NIR lights with different wavelengths but the same intensity of 26.7 W/cm^2^. (**b**) The EPSCs response to 20 light pulses with different wavelengths.

**Figure 9 materials-15-08247-f009:**
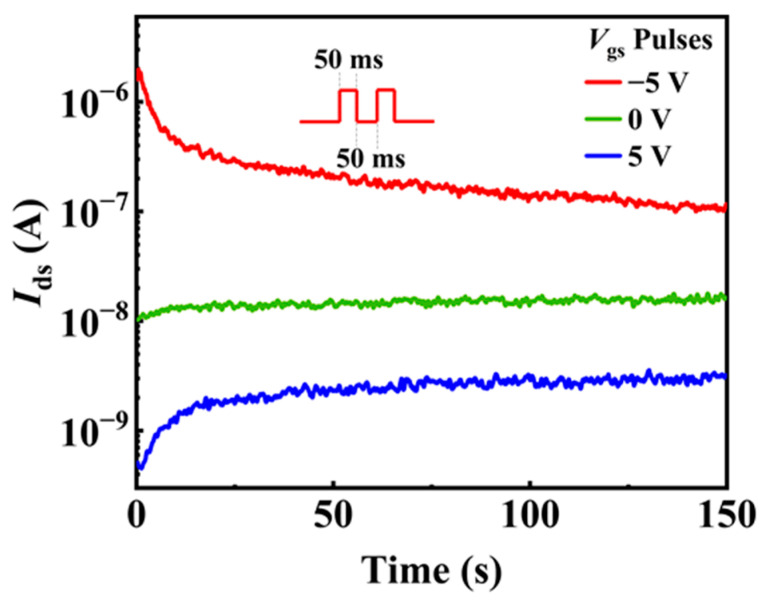
The current *I*_ds_ change with time in the P(VDF-TrFE)-coated InAs NW device without applying gate voltage pulsed (green line), after applying 10 negative gate voltage pulses (red line) and after applying 10 positive gate voltage pulses (blue line). The duration time and interval time of the pulses are all 50 ms.

**Figure 10 materials-15-08247-f010:**
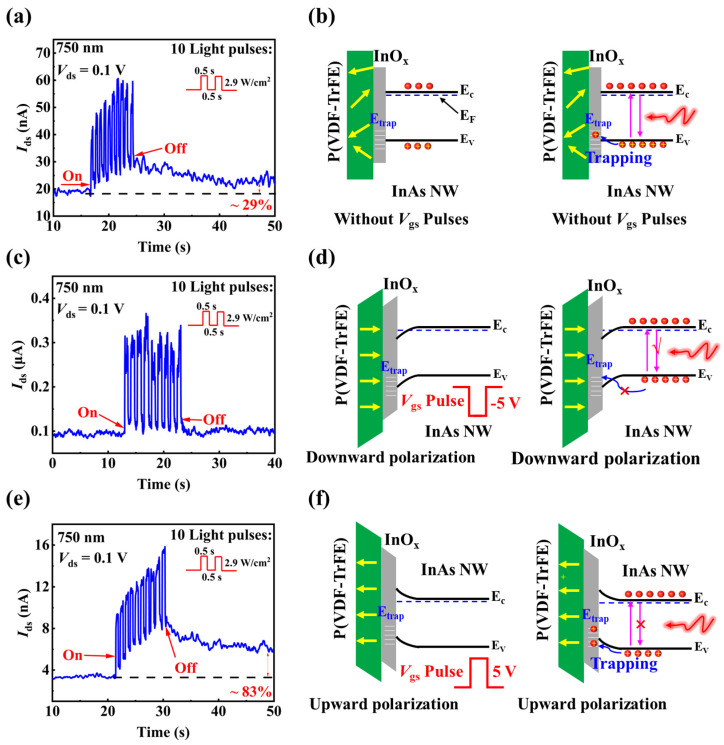
(**a**) The EPSCs response to the stimulation of 10 consecutive light pulses for the P(VDF-TrFE)-coated InAs NW FET before applying *V*_gs_ pulses. (**b**) The energy band diagrams of the P(VDF-TrFE)-coated InAs NW FET before applying *V*_gs_ pulses. (**c**) The EPSCs response to the stimulation of 10 consecutive light pulses for the P(VDF-TrFE)-coated InAs NW FET after applying negative *V*_gs_ pulses. (**d**) The energy band diagrams of the P(VDF-TrFE)-coated InAs NW FET after applying negative *V*_gs_ pulses. (**e**) The EPSCs response to the stimulation of 10 consecutive light pulses for the P(VDF-TrFE)-coated InAs NW FET after applying positive *V*_gs_ pulses. (**f**) The energy band diagrams of the P(VDF-TrFE)-coated InAs NW FET after applying positive *V*_gs_ pulses. In (**b**,**d**,**f**), the left and right images are the diagrams before and after light illumination, respectively. The yellow arrows in (**b**,**d**,**f**) represent the residual polarization direction in P(VDF-TrFE).

**Table 1 materials-15-08247-t001:** A summary of recently published NIR light synapses.

Type	Active Materials	*V*_ds_ (V)	*I*_on_/*I*_off_ Ratio	Response Wavelength (nm)	*R* (A/W)	Features *	PPF Index	Ref.
Organic	P(IID-BT)	−30	3 × 10^3^	550, 850	—	EPSC, PPF, STP/LTP	200%	[19]
	PEA_2_SnI_4_/Y6	40	—	300–1000	200	EPSC/IPSC, PPF, STM/LTM	160%	[38]
	PDPPBTT	−5	~10^3^	808	—	EPSC, PPF	~155%	[39]
	C8-BTBT/F_16_CuPc	0.2	—	380, 640, 790	—	PSC, PPF/PPD, STP/LTP	100%	[40]
	Pentacene	−30	—	790	—	EPSC, PPF/PPD, STP/LTP	~150%	[41]
Inorganic	Si nanocrystals	5	—	375–1342	—	STP/STD, LTP/LTD, STDP	190%	[27]
	Si nanocrystals	0.5	—	532, 1342, 1870	—	EPSC, PPF, STP, STDP	149%	[28]
	ZnO/PbS QDs	0.1	—	980	—	PSC, PPD/PPF, SRDP	14%	[62]
	α-In_2_Se_3_	0.1	—	650–1800	—	EPSC, PPF, STP/LTP	128%	[17]
	α-In_2_Se_3_	0.3	>10^4^	900	—	PPF, STP/LTP	—	[18]
	MoSe_2_/Bi_2_Se_3_/PMMA	−30	—	580–860	—	PPF/PPD, STP, LTP	33.1%	[23]
	MoSe_2_/Bi_2_Se_3_ nanosheets	0.1	—	790	—	EPSC/IPSC, PPF/PPD, STP/LTP	33.7%	[64]
	ITO/Zn_2_SnO_4_/ITO	0.1	—	400—800	0.52 × 10^−6^	EPSC, PPF, LTP	160%	[63]
	Graphene oxide	1	—	365—1550	0.9	EPSC, SIDP/SNDP, PPF, STP/LTP	114%	[30]
	Titanium trisulfide (TiS_3_)	0.2	~4 × 10^2^	400–800	—	STDP	—	[35]
	SWCNT	−0.5	—	520–1310	—	EPSC, LTP	200%	[61]
	InAs nanowire	0.1	6 × 10^3^	750–1550	839.3	EPSC, PPF, STP/LTP	160%	This work

* PPF/PPD: paired-pulse facilitation/depression; SIDP: spike-intensity dependent plasticity; SRDP: spike-rate-dependent plasticity; SNDP: spike-number dependent plasticity.

## Data Availability

Not applicable.

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
