# Peer review of "Near-Infrared Artificial Optical Synapse Based on the P(VDF-TrFE)-Coated InAs Nanowire Field-Effect Transistor"

_materials, 2022, doi:10.3390/ma15228247_

Round 1
Reviewer 1 Report
The topic of the paper is very important. The scaling down to the nanometer range of sensing FET devices is the front edge activity and the research group seems to be able to carry on the research in this field. Therefore, the paper can be the focus of many other groups in the world. However, the layout and the design of the paper lack elaboration of information in many aspects. The comments are as follows.
1. The strongest weakness of the paper is that some of the material is proposed separately from the text of the article under the separate chapter “Supplementary materials”. It is fully not understandable, why this form of the layout has been chosen because it makes the evaluation of the paper extremely irrational. From my point of view, I even pondered to suggest rejecting the article in this layout form fully. So, please elaborate on the whole composition of the paper and do not use the material in the text of the article, which is presented elsewhere.
2. The quality of references is up to date and relatively well balanced. However, the ratio of research results published by non-Asian origin research groups gives a bit tilted picture of the research field as a whole.
3. On page 2, row 86 introduces figure S1, not presented in this paper. A similar situation is seen on page 3, row 109, on page 4, row 141, and later. So, please look at my first comment and make the proper conclusions.
4. On page 3, row 114, introduces the field effect mobility value 1116cm2/Vs calculated from the measured transfer curve. How is the transfer curve measured and what kind of physics is behind the mobility? How we must interpret and understand such an exact value of mobility? Please elaborate on the text and give an additional explanation.
5. On page 4, rows 127 and 128, it is stated that “…the relationship can be fitted by ….”. What kind of new information does relationship tell us? The possible physical mechanisms behind are enough adequately named beyond this sentence.
6. On page 4, row 146, the sweep interval is introduced. Where this output in Figure 2 is seen? Comments are needed.
7. On page 4, rows 155 and 156, the intext formula for responsivity R is introduced. Please refer to the origin of the formula. And additional explanatory comments are strongly suggested as well.
8. On page 4, row 162, introduces the value of maximum responsivity value of 839.3 A/W. What physical condition gives us such an exact value?
9. On page 6 the explanation of PPF behavior is presented. Deeper elaboration is needed.
10. On page 6, row 208, the ratio for describing the PPF behavior. How is this ratio reached? An additional explanation is needed.
11. On page 6, row 211, introduces the light intensity value of 26.7 W/cm2. How has such an exact number been developed? Explanation, please!
12. On page 6, equation (1) for the PPF index must be referred to because it seems not to be the original equation from the authors of the paper.
13. On page 6, rows 214-216, the fast and slow phases are discussed. Why do we need to mix the phases? Please add the explanation.
14. On page 6, row 215, introduces the PPF index value of 160.1%. How much and the exact value is justified and why do we need such an exact value?
15. On page 8, row 295, it introduces a number in percentages (about 29%). Where is the reference to what this percentage is subordinated? An additional explanation must be added.
Conclusion. Important paper on the front-edge topic. Unfortunately, the composition needs further upgrades and corrections before publishing the paper in MDI Materials series.
Reviewer 2 Report
The comments and suggestions are given below for the paper titled: Near-Infrared Artificial Optical Synapse Based on P(VDF- 2TrFE)-Coated InAs Nanowire Field-Effect Transistor
In the presented paper, the authors demonstrate a near-infrared (NIR) (750 to 1550 nm) optical synapse for the first time based on a poly(vinylidene fluoride-trifluoroethylene) (P(VDF-TrFE))-coated InAs NW field-effect transistor (FET). The responsivity of the P(VDF-TrFE)-coated InAs NW FET reaches 839.3 A/W for 750 nm laser illumination, demonstrating the advantage of P(VDF -TrFE) coverage. The obtained results seem interesting in the field of optoelectronics. The paper is well writing and free from scientific mistakes. However, there are some issues that should be addressed in order to improve the manuscript. The main comments are listed below:
1- Some minor writing typos should be corrected.
2- The benefits of FET-based IR photosensors as building blocks for the emerging IR optical communication systems have not been discussed suitably in the manuscript. In this regards, please review the following papers and provide some comments about this aspect in the Introduction Section: https://doi.org/10.1016/j.spmi.2022.107187; https://doi.org/10.1016/j.jallcom.2020.158242.
3. Other interesting Figures of Merit such as Ion/Ioff (noise) should be discussed. Moreover, what about the compatibility of the proposed structure with CMOS platform, which is very interesting for the integration processes.
4. I suggest adding a comparison table including other published results, in order to show the merits of the proposed IR sensor.
In summary, the achieved work showcases significant contributions. Thus, I would recommend with Minor revision this manuscript for the possible publication in the journal.
